# Exploring Environmental Health Inequalities: A Scientometric Analysis of Global Research Trends (1970–2020)

**DOI:** 10.3390/ijerph19127394

**Published:** 2022-06-16

**Authors:** Sida Zhuang, Gabriele Bolte, Tobia Lakes

**Affiliations:** 1Geoinformation Science Lab, Geography Department, Humboldt Universität zu Berlin, 10099 Berlin, Germany; tobia.lakes@hu-berlin.de; 2Department of Social Epidemiology, Institute of Public Health and Nursing Research, University of Bremen, 28359 Bremen, Germany; gabriele.bolte@uni-bremen.de

**Keywords:** environmental health inequalities, health disparities, environmental justice, scientometric analysis, research trends, framework, health determinants, indicators

## Abstract

Environmental health inequalities (EHI), understood as differences in environmental health factors and in health outcomes caused by environmental conditions, are studied by a wide range of disciplines. This results in challenges to both synthesizing key knowledge domains of the field. This study aims to uncover the global research status and trends in EHI research, and to derive a conceptual framework for the underlying mechanisms of EHI. In total, 12,320 EHI publications were compiled from the Web of Science core collection from 1970 to 2020. Scientometric analysis was adopted to characterize the research activity, distribution, focus, and trends. Content analysis was conducted for the highlight work identified from network analysis. Keyword co-occurrence and cluster analysis were applied to identify the knowledge domain and develop the EHI framework. The results show that there has been a steady increase in numbers of EHI publications, active journals, and involved disciplines, countries, and institutions since the 2000s, with marked differences between countries in the number of published articles and active institutions. In the recent decade, environment-related disciplines have gained importance in addition to social and health sciences. This study proposes a framework to conceptualize the multi-facetted issues in EHI research referring to existing key concepts.

## 1. Introduction

The state of the environment is the central foundation for human health and wellbeing. As first defined by WHO in 1993, environmental health comprises those aspects of human health (including quality of life) that are determined by physical, biological, chemical, social, and psychosocial factors in the environment. However, these factors are not equally distributed across a population, as higher levels of exposure to environmental hazards are often found to be associated with unfavorable socioeconomic status and demographic factors [1,2,3]. Such differences in exposure to environmental salutogenic and pathogenic factors can translate into marked health disparities, referred to as environmental health inequalities (EHI) [4]. While “inequalities” is used here to represent general differences in health, many of these differences (particularly where they are linked to social determinants of health) are represented as “inequities” because they are unfair and unjust. “Environmental justice”, another important term in environmental health studies, focuses on a fair and equitable distribution of environmental hazards and benefits within society, and also equal treatment and involvement of all population groups in environmental decision-making [5]. “Health disparities”, on the other hand, may be used interchangeably with “health inequalities” without necessarily implying the presence of injustice [6]. These terminologies, yet different, are all considered to be addressing issues concerning EHI.

According to WHO Regional Office for Europe [5], EHI studies focus not only on differences in environmental health risk factors, but also differences in health status caused by environmental conditions. Risk factors here were interpreted as any attribute, characteristic, or exposure that could affect population health. Furthermore, to clarify the scope of different research domains, we further referred to the definitions from European Environmental Agency [7]: “*Environmental hazard* is the occurrence of a natural or human-induced physical event or a physical impact that may cause loss of life, injury or other health effects, e.g., air pollution.” In addition, we consider *environmental benefits***,** which are natural or human-induced resources that may have a positive impact on human health, e.g., green space [8]. *Social vulnerability* is “the propensity or predisposition of people (individuals or a population of a given area) to be negatively affected by external stressors, including environmental health hazards. It could be seen as the combination of sensitivity (or susceptibility to harm) and capacity to avoid, cope with, or adapt to environmental health hazards. Sensitivity is largely driven by age and health, while the ability to cope is linked to socio-economic status, social support available or awareness of risks” [7].

As defined above, EHI has received increasing attention in recent years in research, society, and politics. Although literature on EHI is burgeoning, few have depicted a panorama of research in this field. Existing systematic reviews or meta-analyses often focus on certain aspects of the physical or social environment, for instance, green space [9,10,11,12], air pollution [13,14,15,16,17], noise [18,19,20], socioeconomic status (SES) [1]. However, populations are not exposed to certain factors in isolation; they are affected by all environmental health determinants, which are interconnected and evolve over time and space. Therefore, studies call for more comprehensive understanding on the multi-faceted issues in EHI [21]. Attempts were made by WHO Regional Office for Europe [4,22] to comprehensively monitor EHI with available data for 53 countries in the WHO European region. Nevertheless, efforts on reviews that also incorporate studies from other regions in the world are still in need.

Furthermore, the multifactorial causes of EHI also require a more comprehensive and comparable approach to assess the multiple influences on health, and to understand the underlying mechanism of how these factors together shape health disparities. Summarizing indices are well-established approaches and represent a mathematical combination of variables reflecting the contributing factors for a certain phenomenon. A carefully constructed EHI index could provide a promising tool to measure, assess, and monitor EHI from multiple perspectives and to inform policy-making and interventions in a spatially explicit, transferrable and transparent way [23,24]. However, due to the multi-faceted nature of EHI as well as constraints on data availability, the identification and characterization of dimensions and indicators that are incorporated in an index could suffer strong subjectivity and limited comparability [23]. Other limitations, such as the limited number of review literature and interdisciplinary research, could also contribute to the problem.

Scientometrics, an analytical instrument for evaluating a large volume of publications and citations, offers a data-driven manner for a comprehensive analysis and domain understanding [25]. Only a few scientometric and bibliometric works on the topic of environmental health have been conducted, and without emphasis on health inequality [26,27], or without highlighting the environmental dimension [6]. Other studies relevant to this topic have a broader scope measuring trends in health research [28], or focus on only certain domains of environment, such as rural environment [29], greenspace [30], air pollution [31,32], and climate change [33]. An attempt has been made by Nelson and Grubesic [34] to visualize the scientific progress of environmental justice research using scientometric techniques; however, more comprehensive and in-depth characterization of EHI studies are yet to be conducted.

To this end, this study adopted scientometric analysis to report on the global research status and evolving trends of EHI research. Combined with a content analysis, this study provides researchers, decision-makers, and stakeholders with in-depth understanding of the salient research topics, as well as the dynamics of the field. In addition, co-occurrence keyword analysis presents a comprehensive research tool that may help to explore the contributing factors and underlying mechanisms of EHI. The main objectives of this study are: (1) to provide an overview of EHI research status by depicting the evolution of publication activity and distribution patterns; (2) to identify research trends and prominent knowledge domain in the field; and (3) to propose a conceptual framework of EHI for future studies.

## 2. Materials and Methods

### 2.1. Data Retrieval

The datasets for this study were retrieved from the Web of Science (WoS) core collection, a comprehensive interdisciplinary, bibliographic database consisting of article references from journals, books, proceedings, and more. WoS core collection was chosen for this study because it is considered as the most precise, standardized, and comprehensive source with high quality indexing for scientific exploration and has a significantly better accuracy in its journal classification and research area categorization than other databases [35]. We conducted a topic search (TS), i.e., we searched for terms in Title, Abstract, Author Keywords, and Keywords Plus within the WoS database. According to the EHI focus of this article we consider the combination of environment and health as essential. In addition, the inequality aspect can be captured by terms such as environmental inequality, disparity, inequity, environmental justice, etc. In summary, this results in the following topic search strategy: TS = (environment* AND health AND (“Environmental justice” OR “Environmental injustice” OR disparit* OR inequalit* OR inequit* OR equit* OR equalit*)). The document language is limited to English. To avoid non-traceable papers, the results of the search were further limited to original articles and reviews, as they represent the majority of documents in the results. Veterinary science was also deliberately excluded as this study is focused on human health studies. The search took place on 5 May 2021, so we have decided to analyze EHI studies published until 31 December 2020. In total, 12,320 publications were selected based on these criteria and were downloaded for further analysis. According to the search results, the first paper on EHI as defined in this search was published in 1970. Therefore, the timespan analyzed in this study is from 1970 to 2020.

### 2.2. Analysis and Visualization

Scientometrics employs multiple mathematical and statistical techniques to describe the impact, quality, and knowledge structure, as well as development trends of research in a given field. This study adopted both numeric indicators and network-based analysis methods, such as co-citation analysis and co-word analysis, to reveal the characteristics and dynamics of EHI research.

As shown in the analysis framework (Figure 1), the number of EHI publication per year was first utilized to describe the evolutionary process of the EHI research. Both general quantity and shift of research categories were calculated and visualized using R bibliometrix packages [36]. The number of publications in the top 10 categories was calculated for the time period (1970–2020) according to the classification system of WoS, which classifies journals into approximately 250 subject categories.

Second, the spatial distribution of EHI research, in terms of countries and institutions, was mapped based on the quantity of publications output during the time period using R packages and Tableau. The publication activity of an institution, which was extracted from the affiliation of the author, represents its EHI publication frequency. ≤5, 6–19, and ≥20 publications for each time period were defined as low, middle, and high activity, respectively.

The co-citation network refers to the network formed when every two papers are cited by another document at the same time. The more co-citations two documents have, the stronger the co-citation relation. Documents featuring strong co-citation relationships are deemed as more semantically related. Moreover, with abstracts or terms extracted from references, Latent Semantic Analysis (LSA) can be applied to calculate the semantic similarity among documents for clustering, and log-likelihood algorithm (LLR) can be used to determine the main topics for the clusters [37]. Hence, co-citation references can help to characterize the knowledge structure and reveal main knowledge domains. The software Citespace 5.8.3 [38] was used to conduct co-citation analysis and to visualize the network clusters.

While co-citation network depicts the general research landscape, co-word network can highlight research topics in a more detailed and straightforward manner. Co-word analysis can effectively map the strength of association between groups of keywords by pairwise counting the number of times they appear in the same paper. Likewise, the more frequently every two key words co-occur, the stronger the co-word relation. Pivotal co-word clusters can then be further identified based on the co-word network by calculating similarity matrices and proximity indices [39]. The stronger the co-word relation, the more likely the two words will fit into the same cluster. In this study, co-word analysis and cluster visualization were both conducted in VOSviewer [39].

Additionally, the feature articles identified by the co-citation network were considered for a content analysis, given their potential influence on the EHI research. Furthermore, a general content analysis was performed on co-word clusters, to re-allocate the retrieved information and provide critical review on prominent knowledge domains of EHI [30]. Notably, author keywords were only considered when generating co-word clusters. Keywords-Plus terms were excluded because they are considered to be more broadly descriptive and thus less precise in representing the core content of an article [40].

## 3. Results

### 3.1. Evolution of Publication Activity

The total of 12,320 EHI publications between 1970 and 2020 identified from our search were summarized in Figure 2. The first study on EHI, as defined in our search strategy, was published in 1970; however, it took 30 years before the annual productivity would flourish at an exponential rate. The development stagnated until 2000 when it started to rise steadily, and the last decade has witnessed a significant increase in the number of publications in EHI studies. Accordingly, three-time intervals were differentiated to better explore the past research shifts and future trends: pre–2000s, 2001–2010, and 2011–2020. It is worth mentioning that despite the steady growth in total number of publications in the WoS core collection since 1990s, the proportion of EHI publications to overall WoS publications is also in the trend of increasing, starting from 0.00011% in 1990 up to 0.058% in 2020, with an exponential increase of proportion since 2010.

The quantity of active journals covering EHI studies has also witnessed a significant increase over time, from only one in 1970 to the highest of 880 in 2020. While EHI studies have certainly increased in a variety of fields, one has to keep in mind the overall increase in publications and journals throughout this period. In the pre-2000s and 2000s, the journal with the largest number of publications on EHI research is Social Science & Medicine, followed by American Journal of Public Health and Health & Place, respectively. In the 2010s, International Journal of Environmental Research and Public Health, became the journal with the largest number of publications on EHI, while Social Science &Medicine dropped to the third on the list.

Figure 2 also shows that apart from the increasing number of publications and journals, the WoS categories involved in EHI studies experienced a slow but steady increase as well. Up to 2020, 155 research categories have been associated with EHI research. On the other hand, the top 10 categories over the three time periods have taken a smaller proportion then previous (Figure 3). Figure 3 also reveals that there are significant structural changes in WoS categories for EHI research. Between 1970 and 2020, Public Environmental Occupational Health covers almost completely the topic of EHI. Since the 2000s, environmental-related subject categories have developed rapidly, such as Environmental Science and Environmental Studies ranking up to the second and third respectively on the list since 2010s, and, together, accounting for more than 20 percent of all disciplines, whereas Social Science Biomedical moved to a lower ranking. Moreover, Geography, Nursing and Green Sustainable Science Technology first made it into the top 10 list of categories in the 2010s.

### 3.2. Publication Distribution: Countries and Institutions

The USA and the UK have always been the leading force in EHI research, accounting for 57% and 24%, respectively, of the total EHI publications. For interpretation of this result, it has to be considered that only publications in English language were included. Table 1 shows the top 10 countries with most EHI publications from the 1970 onward contributed 96.3% (pre–2000), 78.9% (2000s), and 71.6% (2010s) of the global total EHI research. The top 10 countries are mostly developed countries. Notably, developing countries like China and Brazil have made it into the top 10 list in the recent decade. Especially in China, EHI research has grown rapidly from only 3 before 2000 to 527 in the 2010s and ranked fifth for global EHI publication contributions.

Institutions in the USA take up most places in the top 10 productive list over the whole three periods. In the pre-2000s, EHI research institutions were few and institutions with modest activity were mainly located in major areas of the US and the UK. In the 2000s, only a few institutions with high publishing activity were found in the USA, some European institutions showed a middle degree of publishing activity, and almost all the rest institutions only displayed a low publishing activity. In the 2010s, institutions that participated in EHI research grew significantly and expanded to six continents. However, although publication activity has seen a great increase, high level of EHI research publication activity was mainly concentrated to the USA (Figure 4). It is worth noting that, according to the keyword’s extraction, the USA is also the most frequently targeted area of study, followed by the UK.

### 3.3. Research Trends: Co-Citation Analysis

Figure 5 is the co-citation network map of EHI research, which mainly consists of nine main clusters, with smaller sized clusters being omitted to avoid too much trivial information. The weighted mean silhouette value is 0.9309, which is very close to 1 (= represents a perfect separation from other clusters), indicating a high homogeneity within each cluster and the clustering results is with high reliability. All the clusters are numbered in order according to the size (#0–#8, from large to small).

According to the color of the time axis located at the top of the picture, #7 is the earliest cluster, which has most of the work generated in the 1990–1999 time period. The citing documents in Cluster #7 mainly focus on topic of “Race” in EHI studies. One of the key studies identified by this cluster has raised the issue of environmental justice from a community perspective and talked about how people of color are more endangered by lead in households, pollution in neighborhoods, and hazards in workplace [41]. Brown [42] has also uncovered overwhelming evidence supporting the environmental justice standpoint that environmental hazards are inequitably distributed by social class, and especially race.

Cluster #1 is the next on the timeline, which has a research focus of “Income Inequality”. The work of Lynch et al. [43] has made an effort to elucidate the linkage between aggregate income inequality and individual health through three aspects including individual income, psychosocial environment, and neo-material interpretation, highlighting that health inequalities could result from the differentials in accumulation of exposures, which could trace to the sources in the material world, e.g., public infrastructure. Another representative article of this cluster has conducted a cross-sectional study to investigate associations between income inequality and self-rated health, life expectancy, low birth weight, and age-and-cause-specific mortality among several countries, and strong associations between higher income inequality and greater infant mortality were found [44].

Cluster #3 and Cluster #8 both formed in the 2000–2004 time period, covering topics on “Environmental Justice” and “Social Capital”, respectively. In the Cluster #3, environmental justice issues were studied from multiple perspectives. For example, Williams and Neighbors [45] reviewed the available scientific evidence that linked racism to the elevated rates of hypertension for African Americans and suggested that the environmental racism can shape disparities in health outcomes by limiting socioeconomic opportunities, mobility, access to services, and by creating a stigma of inferiority to cause stress. Morello-Frosch and Jesdale [46] dissected environmental justice issue by studying the impact of racial segregation on differentials in exposure to ambient air pollutants and associated risks of cancer. The results revealed that highly segregated metropolitan areas tend to have the highest air pollutant related cancer risks.

In the meanwhile, Cluster #8 grouped articles related to the “Social Capital” aspect of EHI studies. Representative nodes in the cluster include the work of Pickett and Pearl [47] and Macintyre et al. [48]. Pickett and Pearl evaluated the existing multilevel studies that investigated effects of neighborhood or local area social characteristics on health and believed that social support may play a part in the mechanisms in which people live may influence health. Macintyre et al. also focused on place effects on health by conceptualizing the framework of conducting related studies. The article identified features for the constitution of a healthy neighborhood, which are (1) physical features of the environment; (2) availability of healthy environments at home, work, and play; (3) public or private services; (4) socio-cultural features of a neighborhood; and (5) the reputation of an area.

As we can see, the health effects of neighborhood and community has already been catching attention at this stage, later on in the Cluster #0 “Health related resource”, which encompasses the largest number of articles, the discussion about the role of neighborhood health resources in health inequality kept growing and expanding. Mitchell and Popham [49] asserted that the impact of income deprivation in health is less pronounced in populations living in a greener neighborhood, as pathways through which low socioeconomic position may lead to disease could be modified by access to green space. Larson et al. [50] linked local food environment to health outcomes like obesity, and found out that populations living in low-income, minority, and rural neighborhoods are most often affected by poor access to healthy food whereas the availability of high-fat, unhealthy foods (e.g., fast-food restaurants) tends to be greater, leading to poor diets and higher rates of obesity. Besides food environment, physical activity facilities are also considered as health resources which have great impact on obesity patterns. Gordon-Larsen et al. [51] assessed the disparities in geographic and social distribution of physical activity facilities in a US national cohort and found that lower-SES and high-minority block had less local access to facilities, which in turn was associated with decreased physical activity and increased overweight.

Cluster #6 was generated around the same time period as #0, but with topics centered on “Cumulative Risk Assessment”. Brulle and Pellow [52] pointed out that it is important to understand mechanisms in which social vulnerability and environmental hazards may interconnect to create synergistic or cumulative burdens on health, so that possible resolution of environmental health inequity can be discovered. Recognizing the health impacts of combined exposure to multiple environmental and social stressors, Morello-Frosch et al. [53] synthesized the existing scientific evidence on the cumulative impact of exposure to environmental hazards and called for environmental policy to pay attention to the cumulative health implications of exposures confronted with disadvantaged groups. In this context, many work also made efforts to assess and quantify the combined effects from aggregate exposures to multiple environmental hazards and benefits [54].

Cluster #2, #4, and #5 are the clusters with most recent literature, represented as “Green Space”, “Air Pollution”, and “Food Dessert”, respectively. In Cluster #2, the work of Wolch et al. [55] offers a synthesis of Anglo-American literature on the role of urban green space, especially urban parks, in shaping environmental justice and public health. The study recognizes the health benefit as well as the unequal distribution of green space across social gradient, while also pointing out policy intervention towards green injustice should be a balancing act to avoid gentrification. Another article by Rigolon [56] also discusses the environmental inequities regarding access to urban parks across different socioeconomic and ethnic groups and proposed to identify inequities in parks access in terms of park proximity, area or quality, providing a more targeted strategy to address inequities. “Nature and Health” by Hartig et al. [57] focuses on the physical environment relevant to urban planning, design, and policies, and four possible pathways linking nature to health were proposed in the article, involving air quality, physical activity, social cohesion, and stress reduction. In this cluster, the health effects of urban park availability are one of the more studied aspects of green space related EHI research.

Cluster #4 mainly focuses on air pollution exposures, featured by work of Hajat et al. [58]. They conducted a systematic review on research related to the distribution of air pollutants by SES, the evidence in much of the EHI literature from North America, NZ, Asia, and Africa suggests that low-SES populations are confronted with higher exposures of air pollutants. However, they found the results in European research quite mixed. The inconsistent results could be possibly attributed to different study type (ecological study versus cross-sectional or cohort study), different area level (neighborhood, city district, etc.), and different type of social indicator/dimension adopted in various research [13]. As green space and air pollution are both features in the physical environmental that have been recognized to have great influence on human health, “food desert”, which refers to an area that has limited access to affordable and nutritious food, can be viewed as a feature of urban structure and neighborhood social context and often associated with low SES [59]. Cluster #5 identifies the work of Sallis et al. [60] and Ogden et al. [61] as representative nodes. According to these two studies, unhealthy food environments, with available food often processed and high in sugar and fats, are known contributors to the prevalence of obesity in the U.S., which has no trend of decreasing. Furthermore, they investigated physical inactivity as a global pandemic and assessed the associations between physical activity and environmental attributes, such as walkability, access to park, and public transport.

### 3.4. Knowledge Domain: Keyword Analysis

In order to better understand the keywords in terms of research focus, we applied co-word network analysis to map the co-occurrence relationships of keywords, as it may shed light on the internal composition and main knowledge domain of a field. A total of 22,645 author keywords in the retrieved literature were identified by VOSviewer. Moreover, keywords with a frequency of more than 20 occurrences were included in the analysis, and small clusters, in which the number of keywords is fewer than 30, were merged automatically. Finally, the network identified a total of 468 qualified keywords and classified them into three clusters (green, blue, and red). As shown in Figure 6, each node represents a keyword, and the size of the node indicates the frequency of that keyword. The lines connecting the nodes denote the co-occurrence relationships between the two keywords. Three clusters were identified by different colors, representing the main knowledge domains of EHI research. Therefore, by observing and comprehending the clusters and the high-frequency keyword, we can deepen our understanding in the research main domains and focus.

#### 3.4.1. Cluster 1 (Green): Environmental Hazards and Benefits

‘Environmental justice’, ‘public health’, ‘policy’, ‘air pollution’, ‘exposure’, and ‘climate change’ are the nodes of keywords that are highlighted, suggesting that environmental equality is often discussed in terms of environmental justice, and that unequal exposure to environmental hazards, especially to air pollution, are widely investigated. In addition, environmental benefits, such as ‘green space’, ‘parks’, and ‘green infrastructure’ are also presented in this cluster, which indicates that unequal distribution of environmental benefits are also of great concern in environmental justice studies, for they offer mitigation of exposure to environmental health hazards such as reducing noise, alleviating air pollution, maintaining biodiversity, and relieving heat stress [9].

#### 3.4.2. Cluster 2 (Blue): Social Vulnerability

This is the largest cluster, which is characterized by the terms including ‘children’, ‘mental health’, ‘physical activity’, ‘income’, ‘minority’, and ‘care’, etc. The terms represent multiple aspects of social vulnerability, which implies the inability of certain social groups to withstand the negative impacts of environmental hazards due to their compositional and contextual characteristics [7]. For example, keywords such as ‘children’, ‘women’, ‘adult’, ‘minority’, and ‘adolescent’ in the cluster indicate the demographic features of study objects. Next, ‘income’, ‘poverty’, and ‘education’ are all elements of socioeconomic status. ‘Mental health’ is linked to ‘stress’ in the cluster, they are both health outcomes on one hand, but also a risk factor for other health outcomes on the other hand, as a poor state of mind and psychosocial environment has been linked to increased susceptibility to many physical illness [62]. Similarly, the health outcome ‘obesity’ can also be a risk factor, for it is often linked to other risk factors such as reduced ‘Physical activity’ and poor ‘food environment’ and increase the susceptibility to other diseases, such as colorectal cancer [63]. Further connected to ‘walking’, ‘exercise’, ‘alcohol’, etc., these keywords are all related to health behaviors, which are believed to be of influence on the vulnerability of an individual towards certain health hazards. Moreover, ‘care’ and ‘health services’ suggest the resource and ability for people to cope with health conditions, while lack of access to these services could leave the person more vulnerable when confronting harm.

#### 3.4.3. Cluster 3 (Red): Health Disparities

The last cluster has a strong focus on social inequalities in health outcomes, with the most prominent keywords being ‘mortality’, ‘health disparities’, etc. Interestingly, ‘socioeconomic status’ was represented as a large node in this cluster while ‘social determinants’ was located in the blue cluster. The SES was not only connected to ‘mortality’, but also to various health outcomes such as ‘asthma’, ‘preterm birth’, ‘cardiovascular disease’, ‘hypertension’, ‘diabetes’, ‘cancer’, etc., suggesting that the role of SES in all these health disparities has been extensively studied. The explanation could be that SES is linked to health disparities in so many studies that it has formed a very strong co-keyword relations with these health outcomes, therefore the cluster algorithm, which is based on relation proximity, assigns it to this group instead of the cluster “social vulnerability”. ‘Risk’ and ‘risk factors’ have also frequently occurred in literature. It might relate to the fact that “risk” is used as a general term for an estimate of disease frequency in a population in epidemiology. Another possible indication is that risk analysis and identification of risk factors of health outcomes are important topics in the field to address the health disparities.

### 3.5. The Pathway and Framework of Environmental Health Inequality

Referring to the key concepts from the cluster analysis, a framework that facilitates interpretation of the underlying mechanisms of EHI by illustrating links or pathways between the environment and human health is presented. Previous studies have recognized the Driving force–Pressure–State–Exposure–Effect–Action (DPSEEA) framework to be a useful and more efficient tool for analyzing complex environmental health problems compared to other cause-and–effect frameworks, for it addresses all the levels from economic and social dynamics to environmental response, and human health [64]. Hambling et al. [65] reviewed several existing environmental health frameworks and considered DPEESA to be most suitable for identifying hierarchical environmental health indicators, given the holistic view it promotes. It is also effective and efficient to support various levels of targeting actions to improve adverse environmental health impacts, as it provides many entry points for interventions along the environmental health causal chain from the root causes through to the health effects [66,67]. However, DPSEEA has its shortcomings in representing the complex associations and interactions between the multilevel socio-environmental contexts and health outcome if understood in a simple linear manner [68,69]. Thanks to the advantage of the DPSEEA framework in its flexibility, it can be adapted and modified under different scenarios [70]. In this context, with proper modification, DPSEEA framework can demonstrate the interlinking relationships between multiple drivers, exposures and health effects instead of being simplified into linear links.

This study decided to conceptualize the causal chain and formation process of EHI based on the DPSEEA framework, while also incorporating the result of content analysis from the network clusters, to add the perspectives from vulnerability analysis [7] and the scheme developed by Bolte et al. [2] that links social disparities to environmental health. We further supplemented the concepts and definitions for our framework (see Figure 7). Social vulnerability could be generalized into community and individual levels [2,62]. Community-level vulnerability may be characterized by neighborhood resources (e.g., social capital), community stressors (e.g., crime), and structural factors (e.g., health infrastructure). Individual-level vulnerability is composed of individual stressors (sensitivity) and coping ability. They are both under the influence from community stress, a state of ecological vulnerability that could translate into individual stressors. Here in our framework, community level vulnerability is addressed as *social context*.

#### 3.5.1. Driver and Pressure

The very root cause of EHI lies in the macro socio-economic and cultural level. These fundamental effects on health are attached with great emphasis for their very root position on the causal chain [71]. As the economy develops and population grows, the physical environment has to withstand the pressure inflicted by production, consumption, waste generation etc., as well as their consequent releases into the nature. While politics, culture, and ideology (a set of ideas that an economic or political system is based on, that influences the way people behave) may also have influence over socio-economic and population development, together, they also affect people’s state of living, releasing pressure to the society in the form of immigration, discrimination, etc. For example, poverty index, population density, population growth, percentage of people aged less than 16 years or 65 years, rate of urbanization, and immigration could all be possible indicators of driving force to EHI [67].

In terms of actions targeted at these driving forces to reduce health disparities, health and economic policies, systems, and programs that try to improve the living and working conditions of all and allocate resources more reasonably and equally could be possible solutions. At the pressure level, actions could call for fairer public financing, and allocate more financing to control waste emission and release, invent clean green technology, promote sustainable development and provide more opportunities for events that could improve cultural exchange and understanding.

#### 3.5.2. State: Environmental Hazards and Benefits, Contextual Social Vulnerability

In general, human activities add pressures on physical environment and social context. These pressures then contribute to changes in the state of the physical and social environment. For the physical environment, pressure leads to the existence of environmental hazards and benefits. We include basic elements of environmental health in both natural and built environment, which are also obvious in network analysis. For example, air pollution, waste, noise, climate change (e.g., heat), poor housing conditions, and transportation design are all considered to be environmental hazards, while green space and biodiversity has been included as environmental benefits.

As for the social context, it is also viewed as community-level vulnerability in this study from the perspective of cumulative risk assessment [72]. All the economic and human activities, as well as culture differences cause pressure on society structure, and, thus, bring about certain social context characterized by socio-demographic status, residential segregation, social capital, neighborhood resources (e.g., health care, police, and social services), and psychosocial environment (e.g., crime, violence, and traffic injury). On one hand, evidence suggests that differentials in social context are linked to a broad range of health outcomes regardless of individual SES [73], in that neighborhoods with fewer resources and lower social capital will make individuals more vulnerable in face of environmental hazards, and cause negative psychosocial environment, inflicting more stressors on individuals who live in the neighborhood. On the other hand, the individual vulnerability can also shape the social context since the people can choose to relocate within their capacity [7].

Intervention actions at this stage will be more cost-effective if they focus on environment conservation, alleviating environmental pollution and producing more environmental benefits; and with regard to social context, improving neighborhood living conditions and minimizing residential segregation by providing more equitable resources. In addition, monitoring systems for the state of health inequality determinants (e.g., environmental quality, deprivation index, etc.), which allow for problem identification and for evaluation of intervention effects, are needed.

#### 3.5.3. Exposure: Exposure and Access

Exposures occur when humans come into contact with the health hazards, resulting in potential health effects [66]. In this study, the exposure concept actually embraces both exposures to environmental hazards and to environmental benefits. Evidence suggests that more access to the green space and parks would not only reduce some exposure to environmental hazards, as green space could alleviate noise and air pollution, but also mitigate relevant health conditions [10]. Therefore, it would make more sense to consider contacting with hazards and benefits both in measuring the exposure level. Exposure can be translated as external measurements (e.g., accessibility, concentration, proximity, and duration) or internal dose (at target organ); however, for most studies, external exposure is mainly considered, as human biomonitoring is not always available.

It is worth noting that social vulnerability also has an impact on exposure level and may lead to exposure variations, as it has long been recognized that disadvantaged groups may face a disproportionately high risk for being exposed to environmental hazards [2].

At this point, environmental protection measures are needed to reduce or even eliminate detrimental environmental exposures. Additionally, in order to reduce health inequities, individuals and groups of all socioeconomic groups should be empowered to represent effectively their needs and interests regarding exposures. Participatory decision-making process would be a good solution to the problem that disadvantaged groups often having no voice in policy concerning their own life and health, due to the unequal distribution of socioeconomic resources and social capital.

#### 3.5.4. Effect: Health Outcome Disparities

The exposure and access to environmental hazards and benefits would finally lead to health effects. Health outcomes, such as mortality, obesity, mental health, asthma, self-rated health, HIV, cancer, and cardiovascular disease are among the most studied [1,74,75,76,77]. Inequalities in environmental health outcome are the result of unequal distribution of burdens and resources connected with exposure variations.

In addition, given a certain level of environmental exposure, individual vulnerability may also modify the health effects. Factors such as genetic susceptibility, preexisting medical conditions, vulnerable socio-demographic profile (e.g., children and the elderly), and lifestyle (e.g., smoking, alcohol, unhealthy diet habits, and sedentary lifestyle) are considered as individual sensitivity towards health hazards [78,79,80]. In the meantime, psychosocial factors (e.g., stress), risk awareness, and SES factors suggesting individual’s ability/adaptability to cope with health conditions, have also been regarded as vulnerability factors that could modify health effects [2]. In this way, individual vulnerability has an impact on health, characterizing differential exposure-response function.

Actions at this stage will be more effective if there are systems to collect health data regularly, and there are mechanisms to ensure that these data can be analyzed and applied to develop more effective policies, systems, and programs [71]. Education and training of medical personnel and policy-makers, as well as investment in public awareness on the health effect of different social and environmental determinants, are also essential for better addressing health inequalities.

Important potential influencing factors of EHI identified from the network and content analysis (Table 2) were then categorized into the different dimensions proposed by the framework (Figure 7). The keyword frequency may indicate the potential important dimensions and indicators that should be taken into consideration when addressing EHI issues.

## 4. Discussion

### 4.1. An Overview of EHI Research Status

The results indicate that the research on EHI has flourished since the 2000s and received a significantly growing interest during the last decade. With the research originated from environmental racism studies regarding unfair toxic exposure in the US, the research scope keeps expanding, and major research theme has also shifted toward to the role of social disparities in EHI. It was agreed that populations with lower SES would have higher risk of exposure to environmental hazards. During the last decade, increasing attention has been laid on the impact of environmental hazards and benefits on health disparities, with an exponential rate of increase in work studying environmental determinants of health, such as air pollution, green space, urban parks, and food access.

The fact that the top ten categories have taken a smaller proportion of all EHI publications over the three time periods indicates the increasingly multi-interdisciplinary nature of EHI research and suggests that the topics in the field are more and more diverse and inclusive over time. It is worth noting that Geography, Nursing, and Green Sustainable Science Technology ranked up to the top 10 subject categories during the last decade for the first time. This may suggest that environmental health inequalities were increasingly associated with geographic locations [73], and that more health outcomes were considered; and among the environmental factors, the role of green space and green infrastructure in environmental health inequalities has been increasingly discussed [81].

Based on the analysis of publication distribution, this study reveals that the majority of EHI research literature was from and also targeting at developed countries, especially the United States and the United Kingdom, but at the same time, the share of contributions from Canada, Australia, Europe, and China are also increasing significantly. Especially in China, EHI research has grown rapidly from only 3 before 2000 to 527 in the 2010s and ranked fifth for global EHI publication contributions, indicating a growing concern and effort in dressing the environmental health inequalities and injustice. The observation on the distribution of EHI publication might possibly indicate that countries with stronger economic capacity tend to have more participation in the EHI research. However, there are also more aspects to be considered. For example, researchers in developed countries may have an easier access to the journals (e.g., English language, culture, network, funding, etc.).

### 4.2. The Global Research Trends and Knowledge Domains of EHI

This study investigates the evolving trends and salient knowledge domains of EHI mainly by means of keyword co-occurrence (co-word) analysis and literature co-citation analysis. The co-citation network analysis reveals nine main research domains, which are “Race Matter”, “Income Inequality”, “Social Capital”, “Environmental Justice”, “Health-related Resource”, “Cumulative Risk Assessment”,” Food Dessert”, “Green Space”, and “Air Pollution” (from early to present). Among them, “Health-related Resource”, “Income inequality”, and “Green Space” are the largest clusters with most literature. Starting with race topics, the clustering algorithms objectively group EHI studies, in terms of research focus, from social determinants to more of environmental aspects (e.g., green space, air pollution, and food desert). This is consistent with the long existing major concern over SES and increasing interest in the role of green space in public health research. It is worth mentioning that the clustering and naming of the cluster is based on natural language processing methods including Latent Semantic Analysis (LSA) and log-likelihood algorithm (LLR). Authors have also looked through main works from each cluster to ensure the naming is not too deviate from the center of the topics of these literatures. While the clusters could give us some insights into the research trends of EHI, we should also bear in mind that these terms might not be comprehensive enough to fully cover the core and essence of each article in a cluster. Furthermore, search terms and errors in semantic algorithm analysis could also lead to bias in the results.

Co-word analysis identified three main knowledge domains of EHI research, which are summarized as “environmental hazards and benefits”, “social vulnerability”, and “health disparities” based on both keyword frequency and content analysis. In each cluster, many relevant dimensions of EHI have been identified (Figure 6 and Table 2). One interpretation could be that it is the combination of differentials in “social vulnerability” and exposure to “environmental hazards and benefits” that leads to impact on health, and thus the “health disparities” [7]. While the co-word analysis could help identify hotspot topics and knowledge domains of EHI research, the complexities in EHI studies determine, to a certain degree, that the interpretation is inevitably subject to different conceptualization and definition. For example, “social vulnerability” has several definitions in different publications. Neil Adger [82] would include exposure differentials in social vulnerability by defining it as the exposure of groups or individuals to stress as a result of social and environmental change. In this study, social vulnerability is just conceptualized as the propensity or predisposition of people to be negatively affected by external stressors without taking exposure differentials into consideration.

From the analysis in this study, the interpretation of clusters could not avoid a certain subjectivity and ambiguity as well. For instance, the clusters from co-word network reveal some interesting connections behind the inexplicit cutting line. That is, the keyword “socioeconomic status” is so much associated, or at least most studied together, with health outcomes, that it was less connected to the keyword ‘social determinants’ as they are grouped in different clusters despite the fact that socioeconomic status is one of the social determinants of health. Another interesting result is that, according to the keyword frequency, though there is plenty of research on different determinants of health, the efforts for index building are still rare. Many such studies have been carried out in North America and the UK [83,84,85], while in most other countries the topic has not drawn enough attention. One reason for few efforts of index building up to now might be that it does not make much sense when masking interactions (e.g., synergistic effects or antagonistic effects). However, an index may still serve as a valid tool for capturing combined effects of multiple exposures in terms of additive effects. Researchers and decision-makers could investigate the role of a composite EHI index in evaluating EHI based on the specific situations.

### 4.3. Perspectives on Potential Mechanisms of EHI

We propose a DPSEEA-based framework to conceptualize the underlying mechanisms of EHI, which aims to give more targeting actions at different stages of the causal chain. The macro social and economic drivers inflict pressure on both our physical and social environment. It will lead to certain state of our physical environment, which is also interpreted as environmental hazards and benefits; and of our social environment, which is considered as community level of social vulnerability (referred to as social context in this study). Social context could translate into individual stressors to influence individual social vulnerability. The combination of social vulnerability (both level) and exposure to environmental health hazards and benefits will give rise to the effect of EHI. In particular, the differences in social vulnerability will also lead to exposure variations and modifications on health effects, and thus contribute to health inequalities.

To take a further look at EHI mechanisms, there is some inevitable ambiguity in defining certain concept in EHI issues that will affect interpretation. As Pearson et al. [86] pointed out, differences in definitions or interpretations of some environmental health variables have led to different evaluations of a given environmental health problem. For instance, it remains equivocal regarding the effects of transportation. Some studies deemed transportation access as beneficial to health concerning the accessibility and connectivity to health infrastructures and services it provides [87]. On the other hand, evidence also supports the negative health effects of transportation, as populations living close to roads may suffer from road traffic-related air pollution and noise [88,89]. Appraisals on blue space are also divided. While Gascon et al. [90] found positive relationships between blue space exposure and mental health and general health status, other studies may consider the threats posed by water body, e.g., drowning [91], lead in drinking water [92], and water-borne diseases, such as cholera [93]. The potential environmental benefits of green space on health are widely accepted. Nevertheless, the inconsistent definitions for “green spaces” and for measurements of green space access add complexity in characterizing the health implications of green spaces [12,94]. For future EHI studies, it is of great importance to precisely define relevant subjects and make efforts to reach consensus on the common concepts.

As we have noted, the study of EHI involves multiple disciplines. Cross-disciplinary studies, weaving environmental sciences, ecology, biology, sociology, psychology, geography, epidemiology, etc., are in need to deepen our understanding on the issue. Additionally, multiple study designs, including experimental, longitudinal, cross-sectional, case-crossover, etc., should be encouraged to comprehensively understand the interlinks between different factors and health [95]. In public health, there have been challenges in evaluating the health effect of interventions because it is unlikely for researchers to directly control the policy influence and study environment. In this context, natural experiments have been increasingly valued as an effective approach for strengthening experimental evidence in epidemiology. A recent review by Crane et al. [96] encouraged the utilization of natural experiments to provide evidence for health effects, especially when it is infeasible or inapplicable to implement planned controlled experimental research designs.

Currently, most of the work measure exposures using variables capturing external exposures (e.g., share of green space, traffic noise level, air pollutant concentrations in the neighborhood), which also put challenges in acquiring the necessary environmental data, especially for studies with larger scale, problems with missing data and incompatible formatting are very common [97,98]. To this end, remote sensing data could be a possible solution to fill some data gaps. Nowadays, various sensors provide free available remote sensed data for large areas with modest to fine resolutions, which offers an explicit spatial perspective to support the retrieval of information for EHI research [99]. Moreover, geo-referenced individual data are also often unavailable due to all kinds of concerns such as privacy. In this case, the way to measure the level of exposure to certain environmental conditions is often equivocal and disputable. Mobile phone data and geo-located social media data could provide a geographical perspective in epidemiological analysis [100,101]. The convergence of the availability of these new data sources, GIS, image processing techniques, and statistical algorithms creates a fertile research environment for EHI studies. Therefore, incorporating indicators derived from new data source into addressing EHI issues could effectively supplement data on some contextual factors and behavioral information for certain environmental exposures, and, thus, allow a more objective and comprehensive understanding on environmental inequality [102,103,104,105,106,107,108].

### 4.4. Strengths and Limitations

The study incorporates scientometric analysis in depicting general picture of EHI research and tries to identify important dimensions in EHI mechanisms based on co-citation network and keyword cluster analysis. It is commonly asserted that strong co-citation relations indicate similar scientific background and themes, and keywords were carefully selected by the authors to properly capture the essence and core message of an article. Therefore, the co-citation network and co-word analysis can quickly grasp the trends and topics of a specific research field [28]. These methods also allow the possibility of quantitative analysis based on a very large amount of literature data and give a certain degree of objectivity in summarizing the trends and knowledge domains. Furthermore, this study establishes a framework with domains and dimensions identified from the scientometric network analysis. The framework absorbs the advantages from the DPSEEA, adds the consideration of social vulnerability and contextual impacts, and incorporates multiple drivers, exposures and health effects instead of simple linear links.

Despite the strengths, there are also several limitations that should be discussed. First, this study may have limited scope of data collection. We only retrieved publications from the WoS core collection database, and other databases, such as Medline, Scopus etc., were not included. Furthermore, the search terms applied in this study is not in an exhausted manner. More words may turn out to be relevant in our study context, for example, “neighborhood, context, residential, housing” could also imply the meaning of environment, and “differences, deprivation, poverty” could also return studies of interest in this study. However, the inclusion of more specific terms would lead to more relevant literature datasets but also more false-positive results, as each added term will also bring in some irrelevant publications to the results. According to the definition and focus of EHI in this study, we consider the combination of environment and health as essential, therefore we have decided the current search would be enough to give a general picture of the study field. Nevertheless, we should bear in mind, when interpreting the results, some relevant studies maybe missed due to the limited search terms. Another criticism of this study is the mere inclusion of English-language articles, which grants advantage in analysis to researchers, institutions, and journals of English-speaking countries, and, thus, may lead to bias in the distribution patterns of EHI publication performance. Finally, there may be some limitations due to the packages and software used for undertaking this analysis. While these are well-established tools, there may be some specific parametrization settings that could be explored in more detailed comparative studies of scientometric methods. Second, this study has proposed a conceptual framework adapted from previous effort to the conceptualization of dimensions generated from keyword co-occurrence analysis. However, the choice of keywords may be subject to the personal preference of authors and, thus, affect the results of analysis to a certain degree. Hence, case studies are in need to test the logic robustness of the framework. Third, although the causal chain of this framework gives an advantage in developing hierarchical indicators, it lacks more detailed information on the interactions between each dimension and indicator. Future research will also be encouraged to further clarify potential interconnections and pathways from more interactive perspectives.

## 5. Conclusions

In this study, we explored the knowledge base, research trends, and critical issues in the field of EHI research from 1970 to 2020 based on scientometric analysis, aiming to provide an important knowledge support for researchers to carry out follow-up research. More importantly, with network cluster analysis and content analysis, we developed a framework to shed some light on the influencing factors, pathways, inter-linkages and mechanisms of EHI.

As our results indicate, the EHI research body is rapidly expanding, with a steady increase in publications, active journals, involved disciplines, countries, and institutions. The large volume of EHI studies and the multi-disciplinary complex nature of the issue presents challenges in synthesizing key knowledge domains of the field and in understanding the underlying mechanisms of the phenomenon. In our study, scientometric analysis allows a comprehensive approach to address the challenges, and to depict the research landscape, emerging trends, and knowledge structure of EHI in a quantitative manner. The presented methods and workflows may provide new insights for other studies as well.

In the future, more consistent conceptualization and consensus on definitions are in need in EHI studies, and more discussions on the development of EHI indicators and indices should be encouraged, to investigate how to properly assess the accumulative health implications of multiple exposures at different levels. This study could contribute as a foundation for future work to systematically develop EHI indicators based on the dimensions in the framework.

## Figures and Tables

**Figure 1 ijerph-19-07394-f001:**
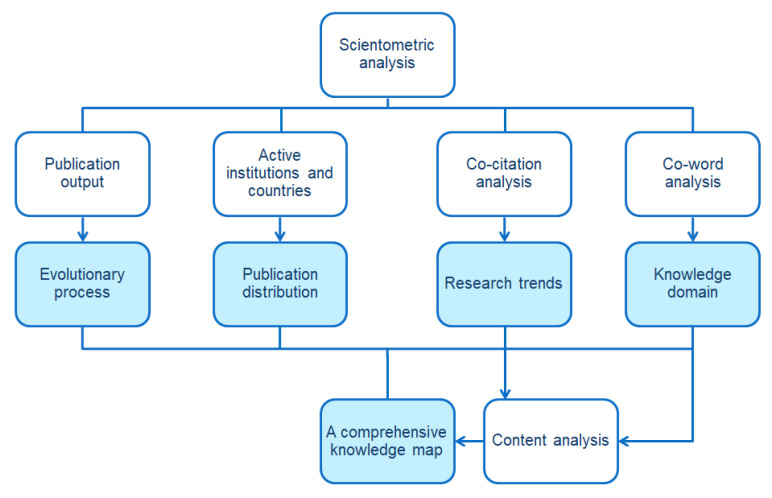
Analysis Framework (white = methods applied, blue shading = results of the analysis).

**Figure 2 ijerph-19-07394-f002:**
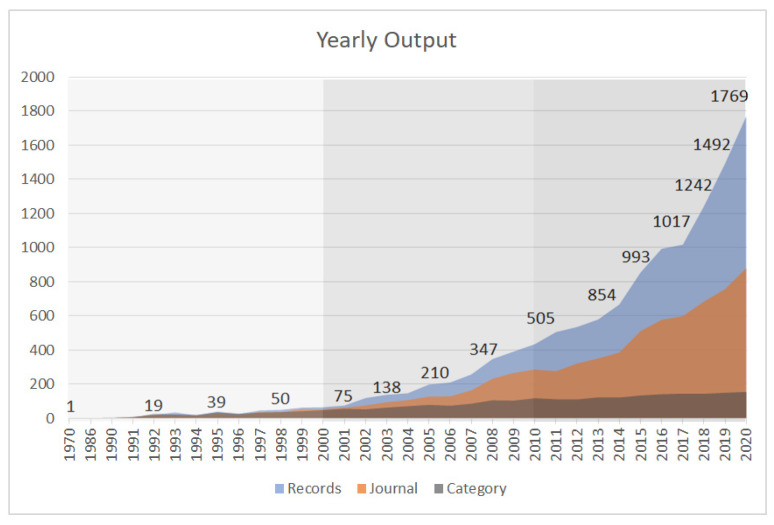
The number of EHI scientific publications, active journals, and categories by year.

**Figure 3 ijerph-19-07394-f003:**
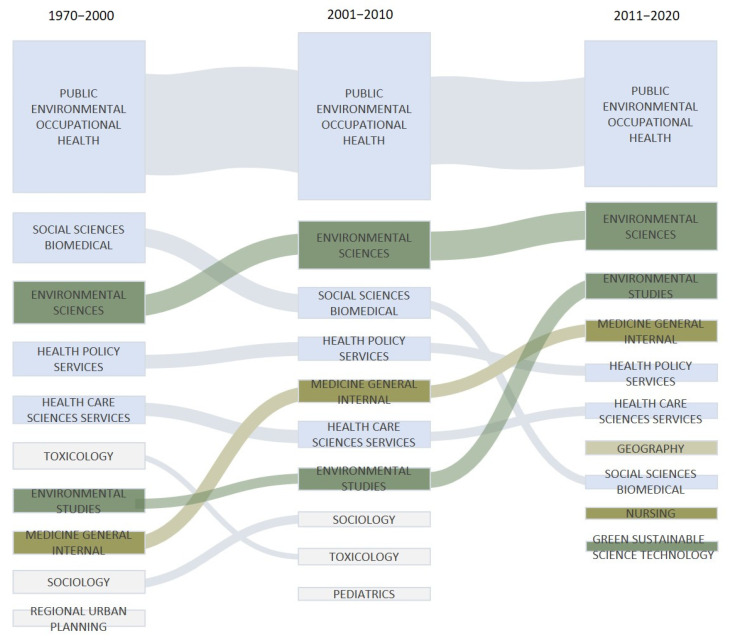
Top 10 WoS Categories of the EHI studies over the three time periods. The percentage of publications that have been associated with each category is proportional to the size of the rectangular.

**Figure 4 ijerph-19-07394-f004:**
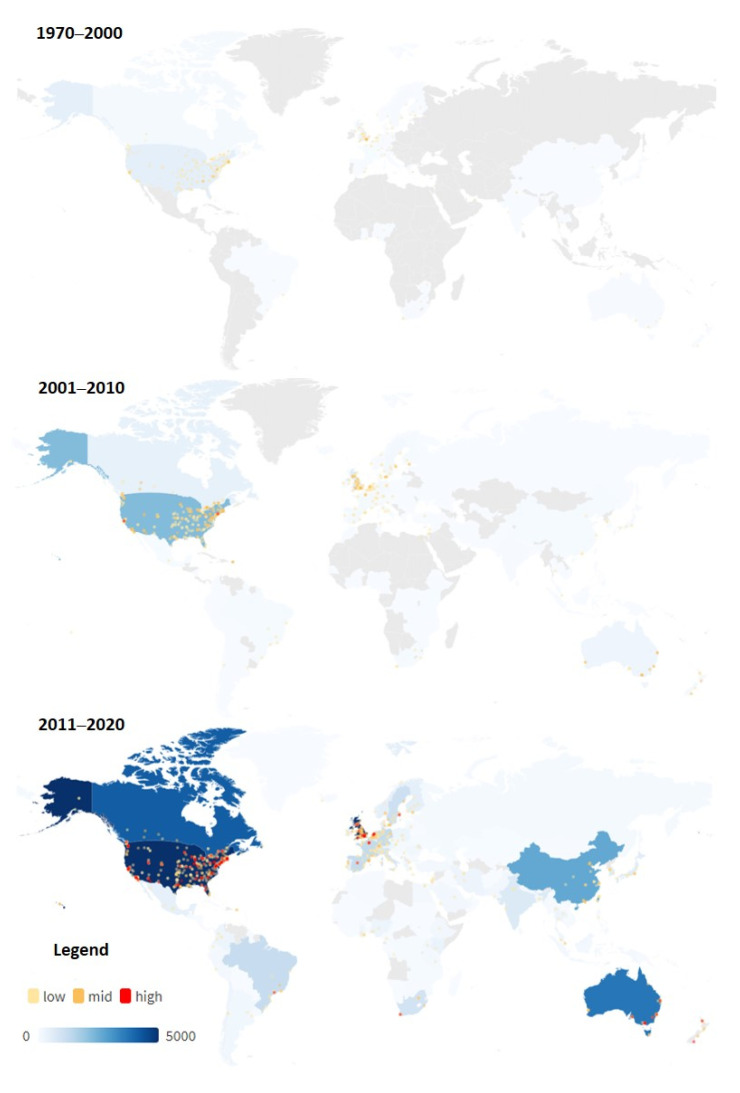
Geographic distribution of EHI publications.

**Figure 5 ijerph-19-07394-f005:**
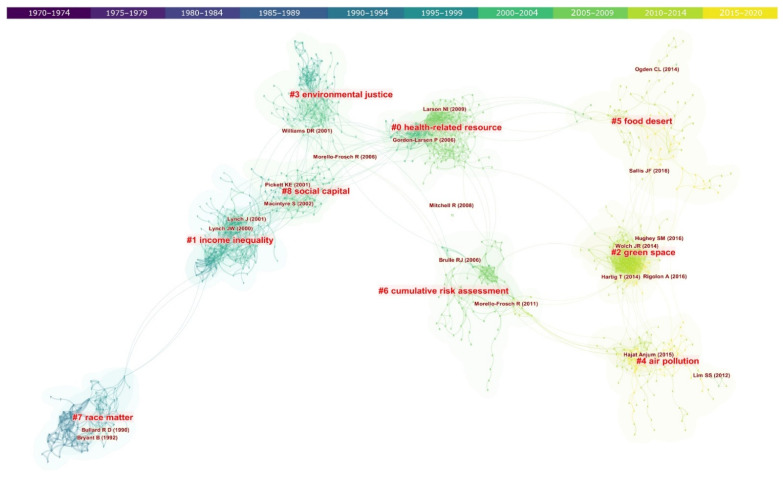
Development of key clusters of EHI publications over time [41,42,43,44,45,46,47,48,49,50,51,52,53,54,55,56,57,58,59,60,61].

**Figure 6 ijerph-19-07394-f006:**
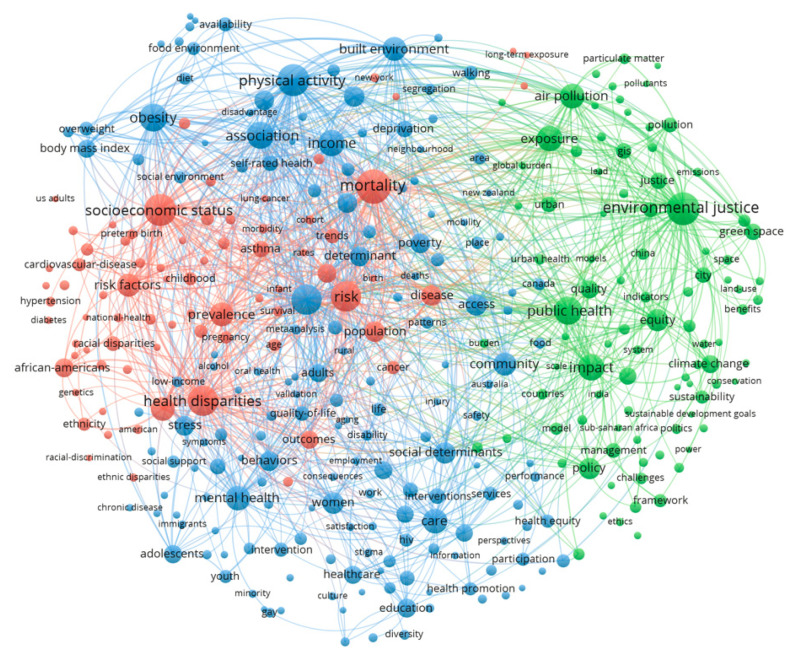
The keyword cluster (green, environmental hazards and benefits; blue, social vulnerability; and red, health disparities).

**Figure 7 ijerph-19-07394-f007:**
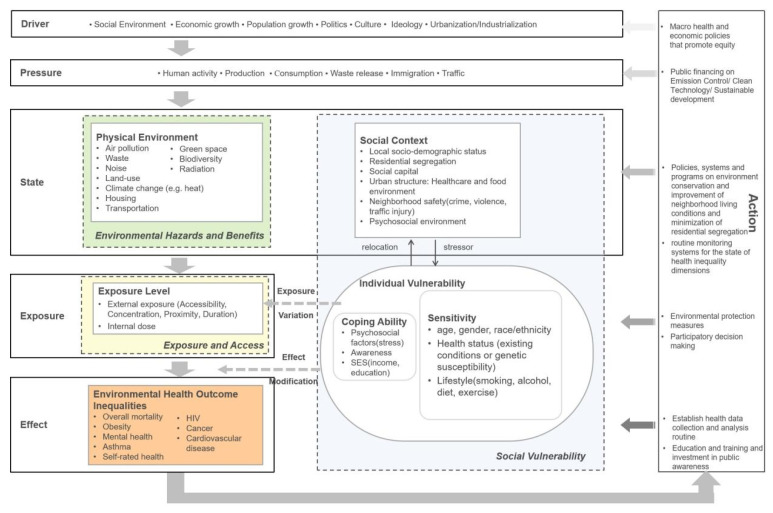
A framework of EHI based on scientometric findings, incorporating concepts from Bolte et al. [2], European Environment Agency [7], Briggs [66], and WHO [67].

**Table 1 ijerph-19-07394-t001:** The top 10 most productive countries of EHI articles.

1970–2000		2001–2010		2011–2020	
Country	Number of Publications	Country	Number of Publications	Country	Number of Publications
United States of America	192	United States of America	1327	United States of America	5478
United Kingdom	87	United Kingdom	368	United Kingdom	2466
Canada	31	Canada	236	Canada	828
Australia	12	Australia	138	Australia	731
Sweden	11	Netherlands	73	People’s Republic of China	527
Netherlands	10	Sweden	63	Netherlands	279
South Africa	9	Germany	49	Germany	251
France	6	France	41	France	248
Brazil	5	New Zealand	38	Spain	243
Denmark	5	Spain	38	Brazil	235

**Table 2 ijerph-19-07394-t002:** Potential influencing factors of EHI based on keyword occurrence under framework.

Main Topic and Subtopic	Number of Publications	%
**Environmental Hazards and Benefits**	3039	24.7
air pollution	894	7.3
built environment	617	5.0
green space	380	3.1
climate change	380	3.1
water	175	1.4
transportation	174	1.4
waste	78	0.6
housing	68	0.6
natural environment	65	0.5
noise	64	0.5
biodiversity	61	0.5
heat	58	0.5
land-use	25	0.2
**Social Context**	1787	14.5
social capital	837	6.8
healthcare	208	1.7
Participation	198	1.6
residential segregation	193	1.6
crime	185	1.5
violence	122	1.0
food environment	91	0.7
psychosocial environment	51	0.4
**Individual vulnerability**	7313	59.4
socioeconomic status	1218	9.9
income	828	6.7
race	825	6.7
gender	448	3.6
education	380	3.1
ethnicity	276	2.2
age	193	1.6
minority	155	1.3
immigration	142	1.2
employment	105	0.9
deprivation	88	0.7
occupation	62	0.5
behaviors	423	3.4
lifestyle	399	3.2
genetics	214	1.7
stress	192	1.6
nutrition	161	1.3
diet	158	1.3
availability	221	1.8
mobility	68	0.6
**Health outcome**	4669	37.9
mortality	1238	10.0
obesity	836	6.8
mental health	648	5.3
asthma	281	2.3
self-rated health	267	2.2
cancer	194	1.6
hiv	190	1.5
cardiovascular-disease	188	1.5
pregnancy	171	1.4
blood-pressure	162	1.3
infant-mortality	113	0.9
birth	113	0.9
breast cancer	88	0.7
chronic disease	61	0.5
injury	61	0.5
psychological distress	58	0.5
**Actions**	1073	8.7
interventions	437	3.5
risk assessment	144	1.2
health policy	100	0.8
urban planning	67	0.5
Index	58	0.5

## Data Availability

Available upon request.

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
