# Peer review of "Exploring Environmental Health Inequalities: A Scientometric Analysis of Global Research Trends (1970–2020)"

_ijerph, 2022, doi:10.3390/ijerph19127394_

Round 1

Reviewer 1 Report

Interesting and revelant scientometric study on Environmental Health Inequalities. Few comments:

* The Discussion is really long and there is much repitition of the Results (e.g. poverty, blue space). It is unfortunate, as the paper is so easy to read until there where it becomes a bit trying. Alternatively have Discussion matters only in Discussion, not in Results (e.g. blue space).

* Add that TS in WOS indicates Topic search (which includes at least title and abstract). Readers, like me, would otherwise wrongly think it stands for "title search" only.

* The inclusion of GDP in Table 1 is a bit confusing. Why is it added? Countries differ in many respects. Perhaps think of addressing any issue with GDP in the text only in the Discussion (where it is now addressed also)?

* ALL figures could be bigger printed, as some are difficult to read / understand.

* Table 5 (twice in the text) should be Table 2?

* Table 2: align left and have first dimension not also underlined.

* Page 22, a bit below half the page (I cannot read line number on print), where it says ".....with most literature. with race". Typo?

Reviewer 2 Report

Dear authors,

This research focuses on the challenges that academics confront when developing critical areas of knowledge in environmental health. The study is significant because it addresses the issue by bringing together the most essential areas of research on environmental health inequalities (EHI) and how to handle them. The research topic is detailed in detail. The author is well-versed in the school's literature on environmental health inequalities (EHI). The study uses scientometric analysis to paint a comprehensive picture of EHI research. It uses a co-citation network and keyword cluster analysis to discover critical aspects of EHI mechanisms. The authors developed a conceptual framework for the EHI mechanisms using scientometric analysis and content analysis. 

The study is well written, contains reliable information, presents new matter, and can be published with minor revisions:

  1. The authors did not explain how 12320 EHI publications were gathered from the Web of Science core collection from 1970 to 2020 in the Methodology section. And why do they primarily choose this period? On the contrary, the authors in the methods section stated that "To build the literature database, references published before 2021 with relevant terms in their titles, abstracts, and keywords were collected," and "the search results were further limited to original articles and reviews."
  2. What software is used in the scientometric analysis? How was it employed in this research?
  3. In the scientometric analysis, what software is used? What role did it play in this study? What are the types of sources including and excluding?
  4. Figure 4. trend in the geographic distribution of EHI publications. It should be Figure 4. Geographic distribution of EHI publications (remove trend in).
  5. Please correct Table 5 to be Table 2 (Page 20, Lane 757) and (Page 22, Lane 837).
  6. I think Table 2 should be at the beginning of the result to present the references: Results (Lane 232). Put this in Lane 243 "After removing duplicates, 12320 documents were finally retrieved from the search between 1970 and 2020 (see Figure 2 and Table 2). The framework (Figure 7) is built based on this table at the end of the results.
  7. Please remove the reference [111] (Page 25, Lane 972]; there are no references in the conclusion. The conclusion may start from "we explored the knowledge base… (Page 25, Lane 972). 
  8. The conclusion needs more explanation about the methods used in the study, the reasons for choosing them, and how they helped reach a new result. It also needs to briefly mention the researcher's findings, their contribution to future studies, and how they will improve future research results. And that future studies are related to the research methods that have already been used. I do not know the relationship between the new data sources, such as remote sensing, mobile data, and social media, combined with diverse study designs, which can offer excellent opportunities to deepen our understanding of EHI.
